# Niche-specific metabolic phenotypes can be used to identify antimicrobial targets in pathogens

Emma M. Glass[1], Lillian R. Dillard[1,2], Glynis L. Kolling[1,3], Andrew S. Warren[4], Jason A. Papin[1,2,3]*

1 Department of Biomedical Engineering, University of Virginia, Charlottesville, Virginia, United States of America, 2 Department of Biochemistry and Molecular Genetics, University of Virginia, Charlottesville, Virginia, United States of America, 3 Division of Infectious Diseases & International Health, Department of Medicine, University of Virginia, Charlottesville, Virginia, United States of America, 4 Biocomplexity Institute and Initiative, University of Virginia, Charlottesville, Virginia, United States of America

* papin@virginia.edu

**Data Availability Statement:** All PATHGENN GENRE models are publicly available on GitHub along with MEMOTE benchmarking scores: https://github.com/emmamglass/PATHGENN. All numerical data to generate figures and scripts used

## Abstract

Bacterial pathogens pose a major risk to human health, leading to tens of millions of deaths annually and significant global economic losses. While bacterial infections are typically treated with antibiotic regimens, there has been a rapid emergence of antimicrobial resistant (AMR) bacterial strains due to antibiotic overuse. Because of this, treatment of infections with traditional antimicrobials has become increasingly difficult, necessitating the development of innovative approaches for deeply understanding pathogen function. To combat issues presented by broad- spectrum antibiotics, the idea of narrow-spectrum antibiotics has been previously proposed and explored. Rather than interrupting universal bacterial cellular processes, narrow-spectrum antibiotics work by targeting specific functions or essential genes in certain species or subgroups of bacteria. Here, we generate a collection of genome-scale metabolic network reconstructions (GENREs) of pathogens through an automated computational pipeline. We used these GENREs to identify subgroups of pathogens that share unique metabolic phenotypes and determined that pathogen physiological niche plays a role in the development of unique metabolic function. For example, we identified several unique metabolic phenotypes specific to stomach pathogens. We identified essential genes unique to stomach pathogens in silico and a corresponding inhibitory compound for a uniquely essential gene. We then validated our in silico predictions with an in vitro microbial growth assay. We demonstrated that the inhibition of a uniquely essential gene, *thyX*, inhibited growth of stomach-specific pathogens exclusively, indicating possible physiological location-specific targeting. This pioneering computational approach could lead to the identification of unique metabolic signatures to inform future targeted, physiological location-specific, antimicrobial therapies, reducing the need for broad-spectrum antibiotics.

for data analysis are available on Zenodo: https://zenodo.org/records/13952471.

**Funding:** This work was supported by the National Science Foundation (GRFP award number 1842490 to EG, NRT-ROL 2021791 to LD, https://www.nsf.gov), the National Institutes of Health (1 T32 GM 145443-1 to EG, R01-AI154242 to JP, R01-AT010253 to JP, https://www.nih.gov), and the National Institute of General Medical Sciences (5T32GM136615-03 to LD, https://www.nigms.nih.gov). The funders had no role in study design, data collection and analysis, decision to publish, or preparation of the manuscript.

**Competing interests:** JP has financial interest in Cerillo, Inc. that manufactured the microplate reader used in some validation experiments.

**Abbreviations:** AMR, antimicrobial resistant; BV-BRC, Bacterial and Viral Bioinformatics Resource Center; CAMP, cationic antimicrobial peptide; FBA, Flux Balance Analysis; GENRE, genome-scale metabolic network reconstruction; KEGG, Kyoto Encyclopedia of Genes and Genomes; MIC, minimum inhibitory concentration; t-SNE, t-distributed stochastic neighbor embedding.

## Introduction

Bacterial pathogens pose a major risk to human health and are responsible for 16% of all global deaths, including 44% of deaths in low-resource countries [1]. Currently, there are over 500 known human-associated bacterial pathogens [2]. Treatment of certain bacterial pathogen infections with traditional antimicrobials has become increasingly difficult in recent years due to growing resistance [3]. It is therefore necessary to gain a deeper understanding of pathogen metabolism to uncover cellular pathways that could be newly exploited for targeted antimicrobial therapies.

Many well-known pathogens have been deeply characterized experimentally and computationally [4–6], but metabolic phenotypes have not been described across pathogen genera. A metabolic phenotype is defined as a unique functional metabolic state of a given organism determined by predicting the distribution of metabolic functions in the network. Systemically capturing complex variation in metabolic phenotypes across pathogens is necessary to begin understanding the development of unique metabolic functions. Understanding variations in metabolic phenotypes will enable identification of unique metabolic signatures in individual or select groups of pathogens, such as those inhabiting the same physiological niche. Identifying physiological location-specific metabolic signatures will allow us to approach antimicrobial drug development differently: we could consider targeting cellular processes that are conserved among pathogens inhabiting a specific body site to reduce the need for broad-spectrum antibiotics, ultimately slowing the emergence of antimicrobial resistance.

To gain a deeper understanding of conserved metabolic signatures in groups of pathogens, we consider classical evolutionary concepts such as natural selection, convergent evolution, and divergent evolution. These phenomena have been previously observed in vitro in a variety of bacterial populations under different environmental conditions [7–10]. Additionally, physiological location has been shown to influence the composition of organisms in the human microbiome through evolutionary pressures. For example, there is a large variation in community composition of commensals present in the skin microbiome across different sites [11], suggesting each physiological location has a unique niche with properties allowing certain commensal bacteria to thrive. Natural selection, convergent evolution, and divergent evolution in bacterial populations are well-studied concepts, and the influence of physiological location on human microbiome composition is well-characterized [7,8,10,11]. However, the idea of physiological location as a selective pressure for the development of unique metabolic function in pathogens has been underexplored.

To capture the range of metabolic phenotypes across pathogens and identify metabolic functions unique to organisms of specific physiological locations, we need to leverage high-throughput, automated, in silico pipelines. Specifically, we use genome-scale metabolic network reconstructions (GENREs) to capture functional metabolism in individual pathogens, at strain-specific resolution [4,12]. Once assembled, GENREs can be used to probe an organism's genotype–phenotype relationship through constraint-based modeling and analysis (COBRA) [13]. In silico modeling of bacterial metabolism through GENREs has proven effective at defining functional metabolism in individual pathogens [4,12,14,15].

Here, we identify unique metabolic signatures of pathogens that share a physiological niche using a collection of 914 GENREs of pathogen metabolism. Additionally, we show that environmental selection pressure is a possible driver of divergent evolution of metabolic function in closely related organisms, while leading to convergent evolution in distantly related organisms that occupy the same physiological niche. Further, we identify genes informed by analysis of the GENREs that are uniquely essential to isolates of a given physiological niche. Finally, we use these predictions of uniquely essential genes to identify and validate possible stomach pathogen-targeted antimicrobial compounds.

## Results

### Models of pathogen metabolism

To sufficiently capture the variation of functional metabolic phenotypes across bacterial pathogens, we generated a collection of 914 in silico network reconstructions of bacterial pathogen metabolism through an automated pipeline (S1 Fig), covering 345 distinct species across 9 phyla (Fig 1A and 1B). The scope and depth of information represented in this collection of reconstructions is substantial: across the collection there are a combined total of >1 million reactions, genes, and metabolites (Fig 1C), with individual reconstructions containing an average of about 1,500 genes, reactions, and metabolites (Fig 1D–1F). The models were constructed using publicly available genome sequences from the Bacterial and Viral Bioinformatics Resource Center (BV-BRC) [16] and paired with open-source software including Python, COBRApy [13], and a recently developed GENRE construction algorithm [17], Reconstructor. We call our collection of metabolic reconstructions PATHGENN, which is the first GENRE collection of all known human bacterial pathogens and is among the largest publicly available collections of high-quality GENREs [18,19].

This collection contains a wealth of metabolic and genomic information, and it was important to validate the quality and relevance of this collection before subsequent analyses. To assess quality of the network reconstructions, we used the MEMOTE score [20]. MEMOTE benchmarking ensures models follow typical community conventions, are in the correct SBML format, and are of high quality. In addition to model format, MEMOTE benchmarks models according to 4 primary areas: annotation, basic tests, biomass reaction, and stoichiometry. Annotation tests ensure that a model is annotated according to community standards: primary annotation identifiers belong to the same namespace and components are described using systems biology ontology terms. Basic model tests ensure that the model is formally correct: there is a presence of genes, reactions, and metabolites within a model. Biomass reaction tests ensure that biomass precursors are present and there is a resulting positive biomass when simulated in silico. Finally, stoichiometry tests ensure biochemical feasibility of the model; if stoichiometric inconsistency is identified, it is likely because of unbalanced reactions. A weighted score is generated for each of these 4 subcategories in addition to an overall MEMOTE score (weighted average of subcategory scores), which allows us to assess the quality of the model generated. The average overall MEMOTE score for reconstructions in the collection is 84% (SD ± 1.06%) (S2 and S3 Figs), suggesting all reconstructions in the collection are of high quality, and therefore biologically plausible. Using the Reconstructor GENRE creation tool in this study allows us to create higher quality models (according to overall MEMOTE score) than other similar reconstruction tools often used in the field. For example, average overall MEMOTE scores for a collection of GENREs created using 2 other automated reconstruction tools (ModelSEED [20] and CarveMe [21]) were reported to be ~40% and ~25%, respectively [17].

Additionally, we determined that the relationship between the number of genes and reactions in the reconstructions in our GENRE collection is logarithmic ($R^2 = 0.973$) (Fig 1G). As the number of genes in a pathogen GENRE increases, the number of reactions in the GENRE increases up to a certain point. However, as the number of genes present continues to increase, there becomes a limit to the number of reactions present. This observation is consistent with the expectation that there are limited evolutionary advantages for bacteria with increasingly large genomes [22], and smaller bacterial genomes are known to have fitness benefits [23]. The trend we see here further validates the physiological relevance of our collection.

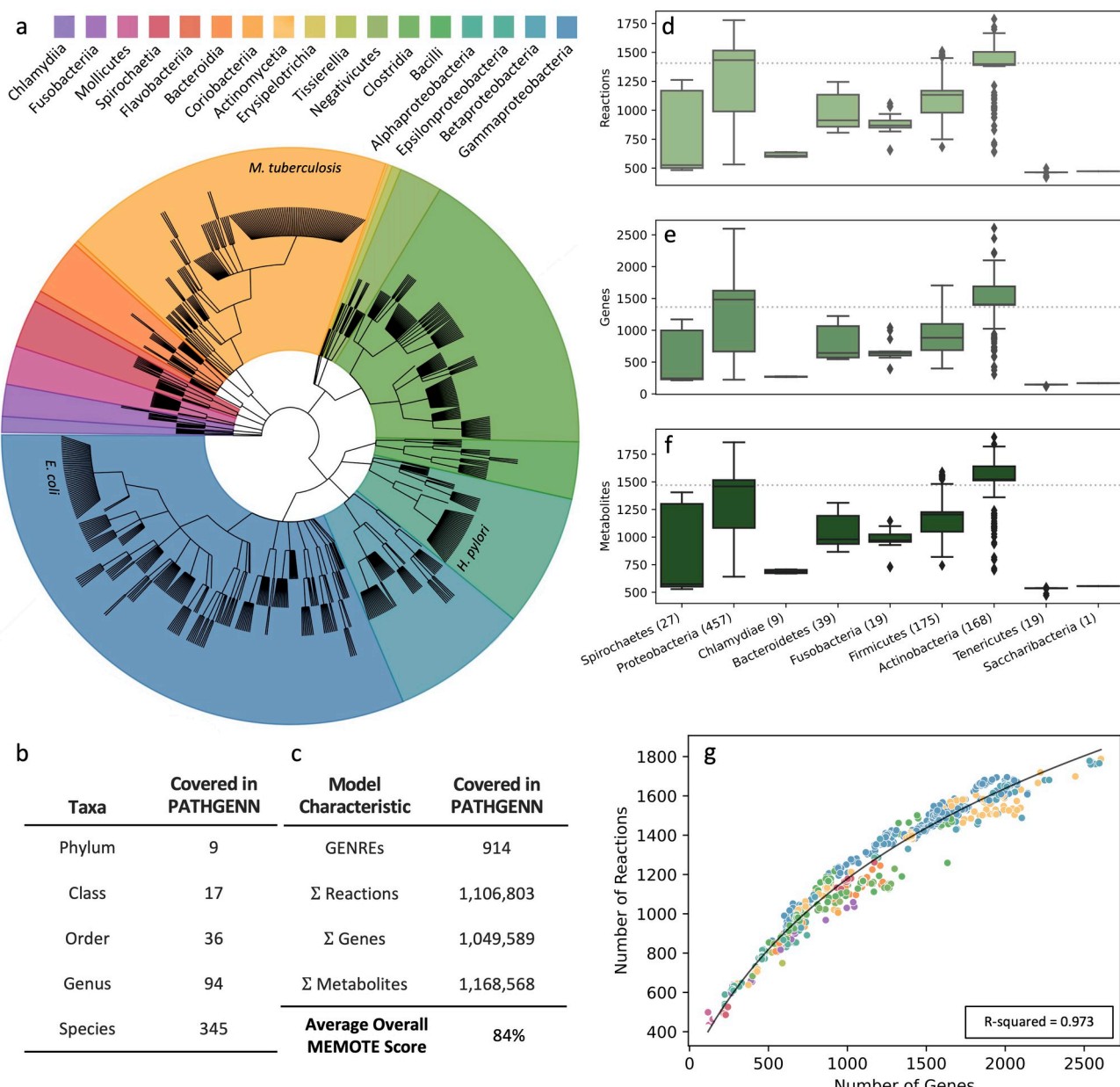

**Fig 1. Scope of our collection of pathogen models of metabolism.** (a) Phylogenetic tree depicting the diversity of 914 considered bacterial pathogens in the collection of GENREs. The many strains of *E. coli*, *H. pylori*, and *M. tuberculosis* (labeled) that are included exhibits the strain specificity of our collection. This cladogram was created using the GraPhlAn python tool. (b) Our collection of GENREs represents 9 phyla, 17 classes, 36 orders, 94 genera, and 345 species of pathogens. (c) Across the 914 models, there are a sum total of 1.11 million reactions, 1.05 million genes, and 1.17 million metabolites. The average MEMOTE score across models is 84% (d–f) Boxplots representing the spread of genes, reactions, and metabolites in each model, classified by phylum. The number in parentheses after the phylum name represents how many models are in that respective phylum. Dotted line in the background represents average Reaction, Gene, and Metabolite numbers across species. (g) The relationship between the number of genes and the number of reactions in each model displays a positive trend similar to other model ensembles. The data presented in 1g was fit to a logarithmic equation. Colors correspond to taxonomic class of pathogen represented by each point (same legend as Fig 1A). The data underlying Fig 1 can be found in Fig 1_Data.zip on Zenodo: https://zenodo.org/records/13952471.

## Identifying metabolic reactions unique to subgroups of pathogens

To exhibit the variety of metabolic reaction subsystems present in all GENREs across the collection, we annotated all reactions according to the Kyoto Encyclopedia of Genes and

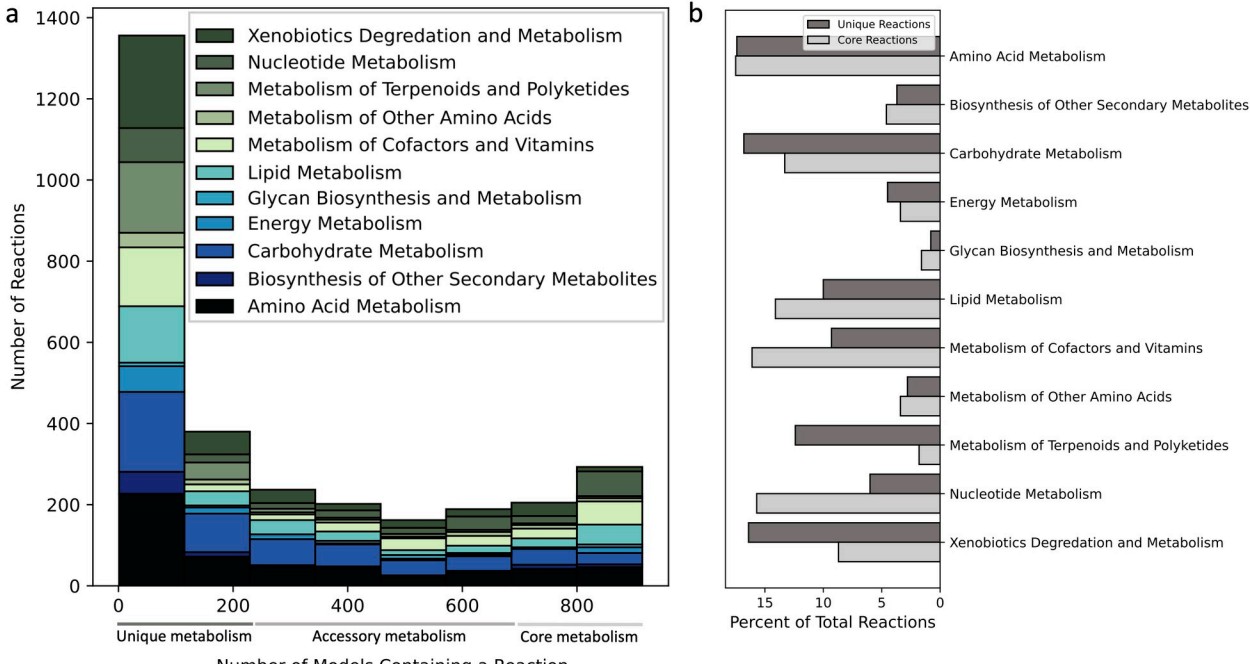

**Fig 2. Core and unique metabolic reaction subsystems across pathogens and the evolution of metabolic function.** (a) Histogram of annotated reactions across models display prevalent reaction classes used in core metabolism (>75% models have a given reaction), accessory metabolism (between 25% and 75%), and unique metabolism (<25%). (b) Relative proportions of metabolic subsystems in core and unique metabolic reaction subgroups. The data underlying Fig 2 can be found in Fig 3_Data.zip on Zenodo: https://zenodo.org/records/13952471.

Genomes (KEGG) and separated reactions into core (present in >75% of GENREs), accessory (between 25% and 75%), and unique (present in <25%) categories (Fig 2A). Details of this analysis are available in the Methods, including rationale for the core, unique, and accessory reaction percentile cutoffs, and a secondary analysis with a stricter cutoff for uniqueness. Furthermore, the analysis presented in Fig 2 allows us to capture metabolic subsystems of both reactions with explicit genetic evidence as well as reactions that were added during the gapfilling process, which goes beyond a traditional bioinformatics-based annotation analysis. On average, 13% of all reactions are gapfilled and 8% of all metabolites are gapfilled across all 914 GENREs (S4 Fig). Gapfilled reactions are added to the models to satisfy the criteria that (1) the metabolic network carries a minimum total flux; and (2) constraints on biomass synthesis and other fluxes are satisfied. These criteria are further outlined in [17]. Consequently, all gapfilled reactions are essential to satisfy these criteria, and most but not necessarily all gapfilled reactions are also essential for biomass synthesis.

Through the reaction annotation analysis presented in Fig 2, we gain a deeper understanding of the distribution of metabolic reactions and their corresponding subsystems across GENREs in our collection. Importantly, we observed that most reactions across pathogens were considered unique (Fig 2A), which can be attributed to the large taxonomic range of pathogens in the GENRE collection. Additionally, we noticed a larger proportion of nucleotide metabolic subsystems in core reactions (10%), which is consistent with the ubiquitous role of nucleotide metabolism across bacterial species [24] (Fig 2B). Furthermore, among the unique reactions, we observed a larger proportion of terpenoid/polyketide (11% more) and xenobiotic (8% more than core) metabolic subsystems. Interestingly, terpenoid/polyketide and xenobiotic reaction subsystems both relate to drug metabolism processes which can be highly variable across

bacteria [25]. Further, xenobiotic pathways are often implicated in antimicrobial resistance [26], suggesting that many pathogens possess unique antimicrobial resistance mechanisms.

Interestingly, focusing only on the leftmost bin in Fig 2A, we observe that most metabolic reactions are present in less than 12.5% of pathogen GENREs. This result suggests that there are certain subgroups of pathogens that share these unique reactions, with the subgroups containing less than 114 pathogens (12.5% of all pathogens in our collection). We can think of these as metabolically unique subgroups of pathogens, because they share unique metabolic reactions that are not present in most pathogens (the other 87.5% of pathogens). Further exploring these unique metabolic subgroups could prove beneficial, allowing us to leverage shared unique functions as possible antimicrobial targets. Identifying unique metabolic subgroups and understanding the evolutionary pressures driving the development of these subgroups is imperative for gaining a deeper understanding of pathogen function.

## Pathogen metabolic function is related to host physiological environment

Our reaction analysis suggested that there are subgroups of pathogens with unique metabolic function. Here, we identify unifying characteristics of these metabolically unique subgroups by examining metabolic phenotypes across our collection of GENREs. To do this, we can generate in silico metabolic phenotypes by predicting metabolic flux distributions that describe the flow of metabolites through a reaction network, ultimately providing us with all feasible metabolic states unique to a GENRE. Metabolic phenotypes are subject to evolutionary pressures like natural selection, which could result in divergent or convergent evolution.

Previous studies have uncovered a strong relationship between in silico metabolic phenotype and evolutionary history, specifically in terms of taxonomic class [19,27,28]. However, earlier investigations have not uncoupled metabolic phenotype from evolutionary history to consider other factors that might play a part in differentiating metabolic phenotypes. Because there has been evidence that physiological location influences human microbiome composition [7,8,10,11], here we investigate the influence of physiological location on the development of unique metabolic phenotypes in pathogens.

We generated in silico metabolic phenotypes of all network reconstructions by utilizing FBA to generate feasible metabolic flux distributions. Metabolic network models are often underdetermined systems of equations with an infinite number of solutions. Rather than trying to calculate a single solution, we perform flux sampling to analyze network properties and determine many potential metabolic phenotypes. The algorithm we use for flux sampling, Gapsplit [29], samples the GENRE's solution space in an efficient way, with the goal of capturing the entire range of possible solutions. To visualize individual metabolic phenotype states in the context of taxonomic class (Fig 3A) and physiological environment (Fig 3B), we used t-distributed stochastic neighbor embedding (t-SNE) (further explained in Methods). Consistent with previous studies [19,27,28], we observe clustering based on taxonomic class (Fig 3A). This result confirms that the flux distributions generated from our GENREs can accurately capture differences in metabolic phenotypes between taxa. Further, we observe clustering based on physiological location, suggesting physiological location may play a part in differentiating metabolic phenotypes beyond taxonomy alone (Fig 3B). Overall, the clustering of metabolic phenotypes in both Fig 3A and 3B suggest that metabolic phenotype is a function of both evolutionary history and environment. Identifying metabolic phenotypes unique to individual physiological niches could be a novel strategy for targeted drug design, so we consider the clustering of metabolic phenotypes in Fig 3B more closely.

We observed several clusters of interest in Fig 3A and 3B, which we considered more deeply for theoretical analysis and biological hypothesis generation. Specifically, we noticed close

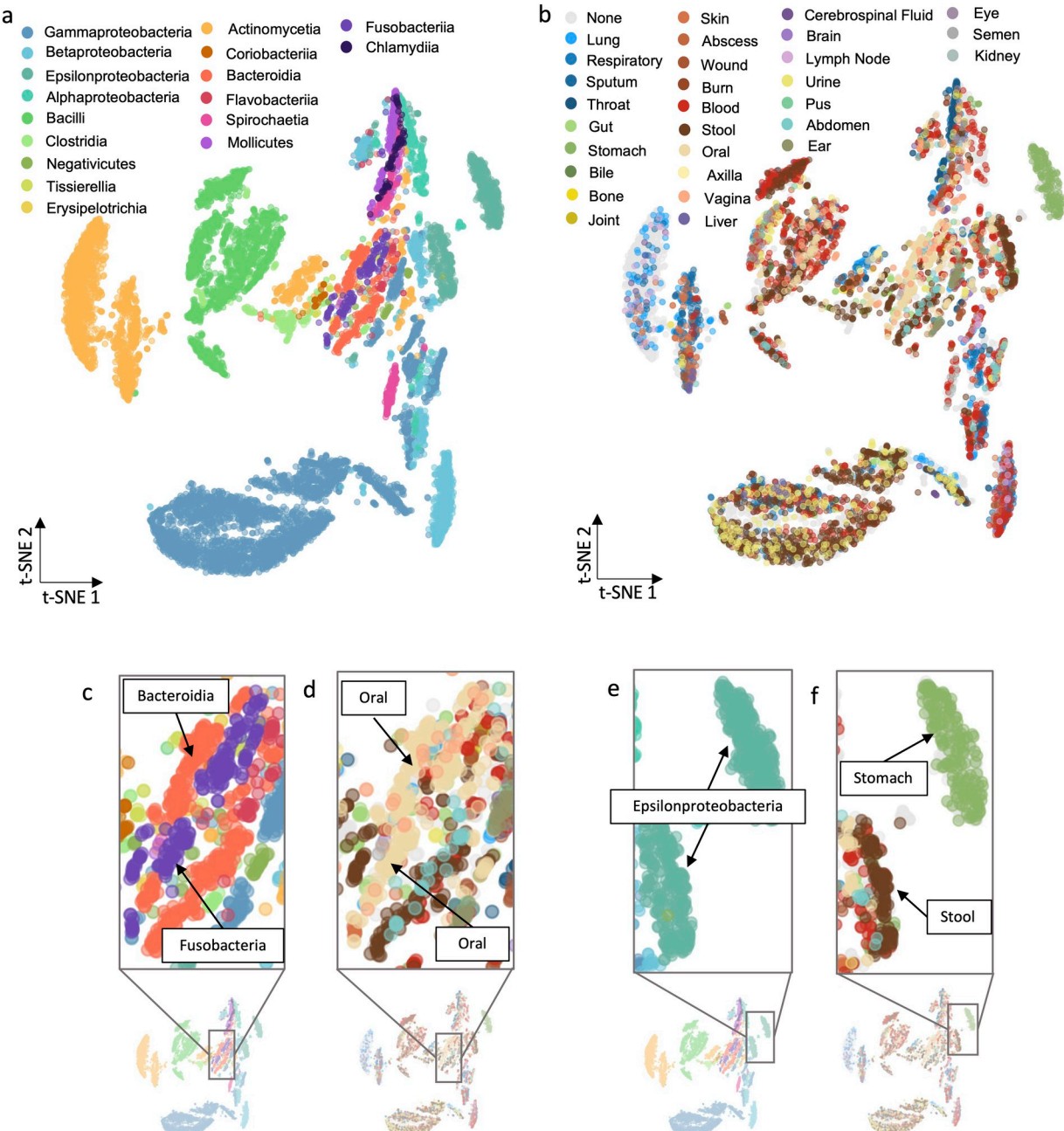

**Fig 3. tSNE of flux samples clustering on taxonomic class and isolation site.** Ten flux samples across all 914 GENREs were plotted using t-SNE, and points were colored on taxonomic class (a) and physiological location (b). Areas of interest from (a) and (b) are indicated in (c–f) and further described in the main text. The data underlying Fig 3 can be found in FluxSampleData.zip on Zenodo: https://zenodo.org/records/13952471.

local clustering of Fusobacteria and Bacteroidia species (Fig 3C). Fusobacteria and Bacteroidia are not genetically similar organisms: a multiple sequence alignment of available 16S rRNA sequences reveals significant evolutionary differences between the groups (S5 Fig). Interestingly, despite having a different taxonomic lineage, these Fusobacteria and Bacteroidia pathogens are all oral-associated pathogens (Fig 3D) suggesting shared metabolic functionality between the groups, as evidenced by the observed clustering. Distantly related pathogens with

shared metabolic phenotypes inhabiting the same environment suggests that unique environmental properties can drive convergent evolution in distantly related organisms.

Secondly, we noticed 2 distinct clusters of Epsilonproteobacteria, more clearly displayed in Fig 3E. The separate clusters suggest that there are 2 distinct metabolic phenotypes exhibited by pathogens of this class. Interestingly, one cluster of Epsilonproteobacteria consisted solely of stomach pathogens (Fig 3F), suggesting that these pathogens have some distinct, unique metabolic functionality compared to their (mostly) stool Epsilonproteobacteria counterparts. This cluster pattern supports the idea that unique physiological environments could play a role in the development of unique metabolic function. Biologically, this result could be explained by the unique physiological environment of the stomach: the high acidity (pH 1.5 to 2.0) [30] allows for only a select few bacterial species to take up residence. Further, it has been shown that *Helicobacter pylori* (a stomach pathogen) has adapted to this extremely unique environment by utilizing metabolic pathways that taxonomically similar organisms do not utilize [31].

The results from our t-SNE cluster analysis support the idea that unique physiological environment could be a driver of divergent evolution in closely related species, while driving convergent evolution in distantly related pathogens. The divergent evolution pattern we observed is of particular interest because it implies that there is a selection pressure for beneficial metabolic functions that are uniquely essential to pathogens in a specific environment. It could be possible to exploit uniquely essential genes of diverse pathogens within a given environment to identify a physiological niche-specific targeted antimicrobial therapy. Therefore, we could approach antimicrobial discovery and repurposing from a different perspective; targeting uniquely essential genes that are conserved across pathogens that inhabit a specific physiological niche.

## The identification of uniquely essential genes informs stomach pathogen-specific antimicrobial targets and growth inhibitors

Here, we leveraged the idea that environment is a major driver for the selection of unique metabolic function to exploit uniquely essential genes of stomach-associated pathogens as potential targets for antimicrobial therapies. Targeting antimicrobial therapies to the site of infection, like the stomach, could potentially ameliorate the harmful effects of long courses of broad-spectrum antibiotics [32]. Additionally, targeting to the infection site could reduce the need for bacterial community characterization since all pathogens in the environment would be targeted. To identify uniquely essential genes to stomach-associated pathogens in silico, we first determined essential genes for all strains in the GENRE collection using an FBA single-gene-knockout method in COBRApy. Subsequently, we instantiated a universal essentiality threshold to determine genes that are uniquely essential across stomach-associated pathogens (see Methods).

We identified 7 genes as uniquely essential to stomach pathogens (and which were not considered uniquely essential in any other physiological environment), *mqo*, *ndk*, *aroE*, *cdh*, *fumC*, *tktA*, and *thyX* (Table 1). Each of these genes meet the defined threshold to be considered uniquely essential genes; however, *thyX* is the most uniquely essential gene with a uniquely essential score of 100% (see Methods). *thyX* belongs to the pyrimidine metabolism pathway, specifically coding for thymidylate synthase. Thymidylate synthase is utilized in both bacterial and human cells and is responsible for catalyzing the DNA building block thymidylate. However, the flavin-dependent thymidylate synthase, *thyX*, is only present in bacteria, and completely absent in humans [33], making it an optimal target for an antimicrobial therapy. There has been evidence of 1,4-napthoquinone derivatives as effective *thyX* inhibitors in *H. pylori* and *Mycobacterium tuberculosis* [33–35], effectively inhibiting bacterial growth.

**Table 1. Genes used for in vitro validation with corresponding targeting compounds and other uniquely essential genes.** We identified 7 uniquely essential genes to stomach associated pathogens (*thyX*, *tktA*, *mqo*, *ndk*, *aroE*, *cdh*, *fumC*) and reported their uniquely essential percentages. Two genes, *fabF* and *fabZ* are not uniquely essential genes, but are used as positive controls in the in vitro validation assay. Gene products, pathways that the gene is involved in are reported for all 9 genes, and the inhibitor, and inhibitor's chemical formula are provided for the 3 genes that were used for in vitro validation and were not researched for genes not used for in vitro validation.

| | Gene Target | Gene Product | Associated Metabolic Pathway | Inhibiting Compound | Compound Formula | % Uniquely Essential |
|---|---|---|---|---|---|---|
| Genes used for in vitro validation | *thyX* | Thymidylate synthase | Pyrimidine metabolism | Lawsone | $C_{10}H_6O_3$ | 100 |
| | *fabF* | 3-hydroxyacyl-dehydratse | Lipid metabolism | Cerulenin | $C_{12}H_{27}NO_3$ | 0 |
| | *fabZ* | 3-oxoacyl-synthase | Lipid metabolism | α-Mangostin | $C_{24}H_{26}O_6$ | 0 |
| Uniquely essential genes NOT used for in vitro validation | *tktA* | Transketolase | Carbohydrate metabolism | N/A | N/A | 90 |
| | *mqo* | Malate dehydrogenase | Carbohydrate metabolism | N/A | N/A | 80 |
| | *ndk* | Nucleoside-diphosphate kinase | Nucleotide metabolism | N/A | N/A | 80 |
| | *aroE* | Shikimate dehydrogenase | Amino acid metabolism | N/A | N/A | 80 |
| | *cdh* | CDP-diacylglycerol pyrophosphatase | Lipid metabolism | N/A | N/A | 80 |
| | *fumC* | Fumarate hydratase | Carbohydrate metabolism | N/A | N/A | 80 |

Further, there has been computational evidence that many versions of 1,4-napthoquinones can inhibit *thyX*, including 2-hydroxy-1,4-napthoquinone, otherwise known as lawsone [33].

This literature review corroborated predictions that *thyX* is an essential gene for bacterial growth and provided us with a potential small molecule inhibitor of *thyX*, lawsone. While our literature review did not verify that *thyX* is uniquely essential to stomach pathogens, it still provided essential insights and promise that there is potential for *thyX* to be uniquely essential and targetable in vitro. It is important to recognize that lawsone has been shown to be effective against non-stomach-associated pathogen species. However, this observation does not discount our hypothesis that *thyX* is uniquely essential to stomach pathogens because of our definition of uniqueness (Methods). A gene can be essential to specific pathogens in each environment without being uniquely essential to pathogens across the environment. For example, 1,4-napthoquinone derivatives (lawsone) were shown to be effective inhibitors of *thyX* in *M. tuberculosis*, a lung pathogen, but that does not mean *thyX* is essential across all lung-associated pathogens. Therefore, it is still possible that *thyX* is uniquely essential and a targetable gene of stomach-associated pathogens. However, due to the unanswered questions that arose during our literature review, and possible un-reported off target effects of this identified inhibitor, it was necessary to validate our computational prediction further.

## Validation of stomach-specific pathogen growth inhibitor

To validate our computational predictions, we designed an in vitro assay to assess growth inhibition of stomach-associated pathogens and non-stomach associated pathogens subject to the possible stomach–pathogen-specific small molecule inhibitor: Lawsone. As previously discussed, there is existing experimental evidence that lawsone is an inhibitor of the gene *thyX* [33,35–39], which we predicted is a uniquely essential gene to stomach pathogens in silico. However, we wanted to validate the potential for lawsone to selectively inhibit growth in stomach-associated pathogens.

Additionally, we selected 2 compounds as positive controls, cerulenin and α-mangostin, which are known to inhibit the *fabF* and *fabZ* genes (not uniquely essential to stomach pathogens). *fabF* and *fabZ* encode acyl-carrier-protein dehydratase and acyl-carrier-protein

synthase, respectively, which are both proteins belonging to the fatty acid biosynthesis II pathway. This pathway has been studied as a target for novel antimicrobial compounds due to it being ubiquitous across bacteria [40]. A previous study exhibited down-regulation of *fabZ* expression in *Staphylococcus epidermidis* in the presence of an α-mangostin inhibitor, and subsequent bacterial growth inhibition was demonstrated. Further, it was reported that the bactericidal action of the α-mangostin inhibitor is comparable with cell membrane lytic cationic antimicrobial peptides (CAMPs), suggesting this compound is a very effective antimicrobial [37]. Further, several prior studies have shown the effectiveness of cerulenin as an inhibitor of *fabF* in vitro and have even characterized the mechanism of inhibition [38,39,41]. In the first study citing cerulenin as a *fabF* inhibitor, cerulenin was shown to be a weak inhibitor of *Escherichia coli fabF* and showed weak growth inhibition, while being a strong inhibitor of *S. aureus fabF* with associated strong growth inhibition [39,42].

We selected 7 total pathogens for our in vitro growth inhibition assay: 3 stomach-associated pathogens (*Arcobacter butzleri*, *Helicobacter pylori*, and *Campylobacter coli*) and 4 non-stomach associated pathogens (*Porphyromonas gingivalis* (oral), *Pseudomonas aeruginosa* (wound), *Escherichia coli* (stool/gut), *Burkholderia cenocepacia* (cystic fibrosis lung)) as controls. Each of the 4 non-stomach isolates have important characteristics that make them desirable candidates for negative controls in this validation experiment. First, each of these non-stomach isolates are gram negative. This is important because the differences in cell wall composition (presence/absence of outer membrane) will not be a confounding factor when validating our computational predictions since all stomach isolates are also gram negative. Secondly, each of the non-stomach isolates is from a different physiological niche which will help support our hypothesis that the selected compounds were selective inhibitors of stomach-associated pathogens. Third, the 4 non-stomach isolates were from varying degrees of taxonomic relatedness to the stomach pathogens. One of the non-stomach pathogens (*P. gingivalis*) is from a different phylum than the stomach isolates, while the other 3 isolates (*P. aeruginosa*, *E. coli*, *B. cenocepacia*) were from the same phylum. This variety in lineage also supports our hypothesis that physiological niche is a driver of metabolic function independent of taxonomy. Finally, the non-stomach isolates were readily accessible and easily culturable in our experimental system. We subjected each of the 7 selected isolates to lawsone, cerulenin, and α-mangostin, that inhibit *thyX*, *fabF*, and *fabZ*, respectively, and continuously monitored growth through stationary phase (further described in Methods).

The *thyX* inhibitor, lawsone, inhibited growth in all stomach-associated isolates (*A. butzleri*, *H. pylori*, *C. coli*) (Fig 4A–4C), while not inhibiting growth of non-stomach associated isolates (*P. gingivalis*, *P. aeruginosa*, *E. coli*, *B. cenocepacia*) (Fig 4D–4G). These results align with our computational predictions that lawsone will inhibit growth of stomach-associated isolates while not affecting growth of non-stomach isolates (7/7 computational predictions correct (100%)).

The *fabZ* inhibitor, α-mangostin, inhibited growth in one of 3 stomach-associated pathogens (*H. pylori*) (Fig 4H–4J). Further, there was no inhibition of growth in *P. aeruginosa*, *E. coli*, and *B. cenocepacia* (Fig 4L-4N). However, there was evidence of growth inhibition in the non-stomach-associated pathogen *P. gingivalis* (Fig 4K). These results could be explained by previously reported off-target effects of α-mangostin. α-mangostin has been implicated in the down-regulation of many genes besides *fabZ*, as well as up-regulation of genes related to oxidative stress [37]. α-mangostin was also implicated in the down-regulation of many other genes, as well as up-regulation of genes related to oxidative stress. The mixed inhibition observed with α-mangostin suggests that it is not a selective inhibitor of *fabZ*. Further, the growth inhibition of *H. pylori* observed in these results suggest that this growth assay can capture previously published data regarding off target effects of certain compounds, as *fabZ* was not considered uniquely essential to stomach pathogens.

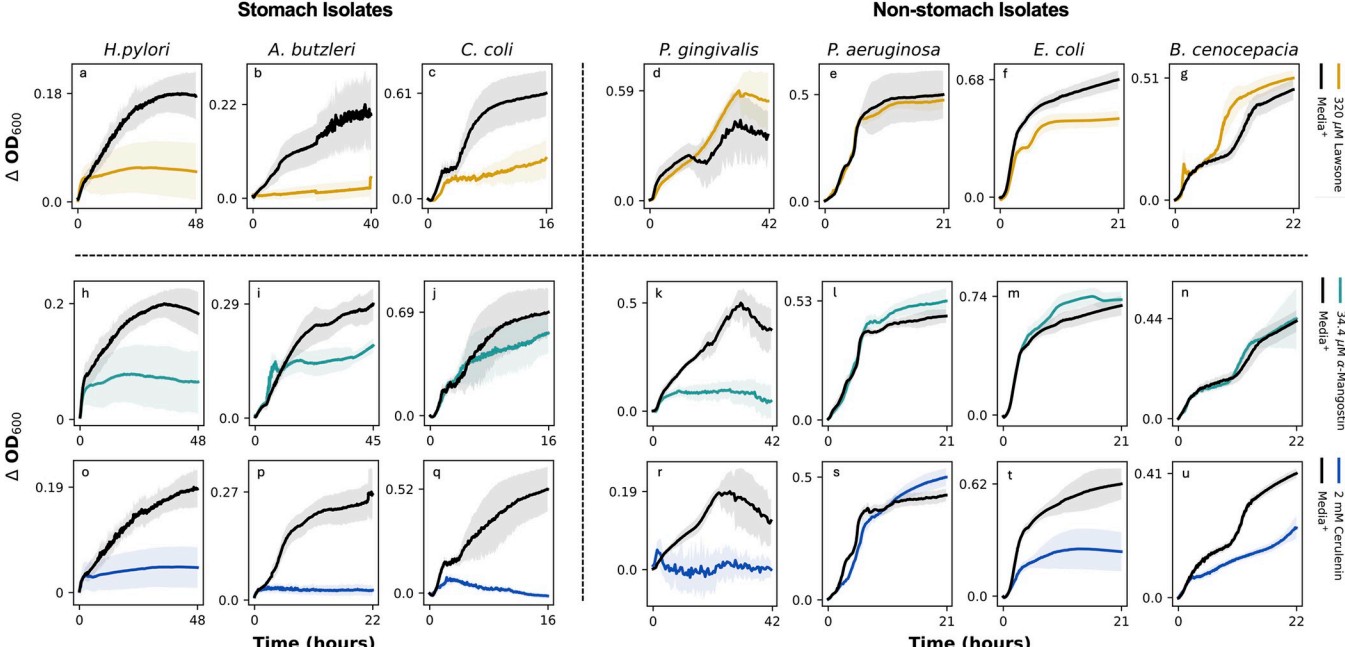

**Fig 4. Results from validation experiments.** Growth *of H. pylori, A. butzleri, C. coli, P. gingivalis, P. aeruginosa, E. coli,* and *B. cenocepecia* in the presence of 3 compounds, lawsone (*thyX* inhibitor, uniquely essential gene inhibitory), α-mangostin (*fabZ* inhibitor, non-uniquely essential control), and cerulenin (*fabF* inhibitor, non-uniquely essential control). Lawsone, top row (a–g). α-mangostin, middle row (h–n). Cerulenin, bottom row (o–u). Black growth curves show the growth of the given bacteria in media without an inhibitory compound. The yellow, teal, and blue growth curves show the growth of the given bacteria in the presence of lawsone, α-mangostin, and cerulenin, respectively. Shaded regions around the growth curves show the standard deviation across 8 technical replicates. The data underlying Fig 4 can be found in Fig 4_Data.zip on Zenodo: https://zenodo.org/records/13952471.

The *fabF* inhibitor, cerulenin, inhibited growth in all 3 stomach-associated isolates (Fig 4O-4Q), while not inhibiting growth in *P. aeruginosa* (Fig 4S). However, cerulenin also showed signs of weak growth inhibition in *E. coli* (Fig 4T) and *B. cenocepacia* (Fig 4U), and strong growth inhibition in *P. gingivalis* (Fig 4R). Importantly, our result that cerulenin is a weak inhibitor of *E. coli* growth concurs with a previously published study [39]. However, unexpected inhibition of stomach isolates suggests the possibility of undocumented off-target effects of cerulenin. The mixed inhibition observed in this test highlights the possibility of this assay to be able to identify and fill gaps in model predictions, resulting in more informed models. For example, we see growth inhibition of stomach isolates subjected to Cerulenin despite *fabF* being a predicted non-essential gene in the models. Given this knowledge, we could modify our models to include *fabF* as an essential gene in future iterations of this project. Overall, the results of this control experiment confirm that cerulenin is also not a selective inhibitor of stomach-specific pathogens.

Additionally, *P. aeruginosa, E. coli,* and *B. cenocepacia* all belong to the proteobacteria class, and all exhibit similar responses to α-mangostin. However, *H. pylori, A. butzleri,* and *C. coli* also all belong to the same class, Epsilonproteobacteria, but exhibit stark differences in responses to α-mangostin. This data further supports our claim that taxonomic class is not the only indicator of metabolic function, and that there are other factors at play driving unique metabolic functionality.

The results validating our computational prediction that lawsone (*thyX* inhibitor) is a stomach-specific pathogen growth inhibitor are particularly encouraging; we inhibited growth of all 3 stomach-associated pathogens while having no effect on growth of non-stomach associated pathogens, suggesting the possibility of stomach pathogen-specific growth inhibition. With these results, we successfully demonstrated that our computational pipeline is valid for

identifying unique antimicrobial targets for growth inhibition of physiological-location specific pathogens. The idea of targeted antimicrobial therapies has been previously explored, usually applied to targeting specific species of bacteria using antimicrobial peptides [43,44]. However, there has been little research on site-specific targeting of antimicrobial compounds. Developing site-specific targeted antimicrobial compounds with a data- and model-driven approach as described here could be a valuable new avenue to explore.

## Discussion

The antimicrobial resistance crisis is rapidly reducing the effectiveness of current drugs to treat microbial infections [3]. There is a need to begin to use creative approaches to identify new or repurposed compounds that can be used as antimicrobials. There is limited work exploring evolutionary pressures that lead to the rise of conserved microbial functions, which could serve as site-specific antimicrobial targets. Here, we leveraged large-scale genomic data and metabolic network modeling to uncover relationships between evolutionary history and unique metabolic function, which we used to identify and validate a physiological niche-specific, targeted, antimicrobial compound.

Our collection of 914 GENREs of pathogen metabolism provides is a valuable resource that can be used in further studies to explore questions related to pathogen metabolism. We acknowledge the quality of our GENREs is limited by (1) annotation quality; and (2) genome sequence quality. During our initial genome sequence selection process, we were unable to stringently select for genomes based on specific completeness and contamination thresholds in the BV-BRC database (beyond the thresholds described in Methods). We acknowledge that more strict completeness and contamination cutoffs would allow our computational predictions including our essential gene analysis to be more robust and rigorous. However, we have ensured that the essential genes predicted for a more complete subset of GENREs are representative of the essential genes predicted for the whole collection (S1 Text). Nonetheless, our collection of GENREs is of high quality according to reported MEMOTE scores (S2 and S3 Figs) and serve as a valuable community resource.

Secondly, when interpreting essential gene results, conditionality is an important factor to consider. Generally, it is known that in both in vitro and in silico computational settings, essential gene predictions will vary based on media condition. Essential gene predictions in a rich media context are more robust predictions and result in a minimal set of essential genes. Alternatively, essential gene analysis in a minimal media produces a larger set of predicted essential genes. This idea has been quantitatively explored in a previous study [45] and showed a linear relationship between the number of essential genes predicted with GENREs and the number of in silico media components: the more components present in the in silico media, the fewer essential genes are predicted. By using a rich media in both our in silico simulations and in our in vitro validation studies, we are left with a minimal set of essential genes that are maximally essential. In future studies, we could consider performing our computational and experimental analyses in the context of multiple physiological niche-specific media conditions. However, we would need to take care to ensure that the results of our analyses were not being influenced or confounded by differences in compounds present in the defined media.

Additionally, we acknowledge the possibility that the stomach pathogen uniquely essential genes could be present in commensal organisms or that the identified inhibitory compound lawsone could inhibit growth of commensal organisms. While generating a large collection of commensal organisms for cross-referencing was outside of the scope of this project, we generated 4 GENREs representative of the most prominent phyla of commensal gut bacteria (Firmicutes: *Enterococcus faecium*, Bacteroides: *Bacteroides intestinalis*, Proteobacteria: *Escherichia*

*coli*, Actinobacteria: *Bifidobacterium longum*). With these 4 GENREs, we generated a list of essential genes in a complete media context. Only one of the 7 uniquely essential stomach pathogen genes (*tktA*) was considered essential in one of the 4 selected commensal isolates (list of all essential genes for the 4 commensals available in S6 Fig). It is encouraging that only one uniquely essential gene was found to be essential in only one of the 4 commensal isolates; but it is difficult to draw conclusions about whether the stomach pathogen uniquely essential genes would be present in all commensal bacteria. Nonetheless, this analysis provides us with evidence that the stomach pathogen uniquely essential genes are not ubiquitously essential in all commensals. While this work provides a first step in developing antimicrobial therapies targeted to the site of infection, we believe that there would be significant value in developing and using a large collection of models of commensal organisms for cross-referencing our computational predictions.

Considering our in vitro validation assay, the 4 non-stomach isolates we selected have favorable characteristics to be negative controls in this study, including: (1) all are gram negative; (2) all are from different physiological locations; (3) 3 were from the same phylum as the stomach isolates; and (4) all were easily accessible and readily culturable in our experimental system. Despite selecting 4 non-stomach isolates with certain key characteristics; there is ample opportunity to expand the scope of our validation experiment in future iterations of this study. This expansion would involve including a larger variety of both stomach pathogen isolates, non-stomach pathogen isolates, and commensal organisms to solidify our conclusion that lawsone is a selective inhibitor of stomach-specific isolates. However, due to the combinatorial issue that is presented when choosing isolates to account for a wide range of phenotypes, for this validation assay we selected the 4 non-stomach isolates detailed above.

Further, there is opportunity to expand the number of uniquely essential stomach–pathogen genes tested in our in vitro validation assay. We selected *thyX* and the corresponding inhibitor lawsone, because *thyX* was the most uniquely essential gene to stomach pathogens. In further iterations of this study, we could identify inhibitory compounds for all 7 essential genes that meet our uniqueness threshold and validate those predictions as well. However, this extended validation was outside the current scope of this study. Further, the selected inhibitory compound (lawsone) does have existing literature describing its mechanisms of inhibition in different bacterial species [33,35]. This literature provides evidence that lawsone does target the gene of interest (*thyX*), but our study does not allow us to indicate or validate the mechanistic driver of drug activity. It is possible that there could be off-target effects of each selected inhibitor (including the positive controls cerulenin and $\alpha$-mangostin), which could be why we see inhibition of some isolates when we did not expect to. To resolve this issue, we could more deeply profile each pathogen and each inhibitor in in future studies.

Our data-driven approach provides a valid framework for identifying highly targetable physiological locations. While our collection of metabolic network reconstructions is of high quality, further curating our model simulations and validation experiments in the ways discussed above would improve this work in future iterations. This computational pipeline and approach is an important beginning step for further discovery and validation of targeted, site-specific antimicrobial compounds that could eventually be brought to a clinical setting to help reduce the harmful effects of broad spectrum antibiotic use.

## Methods

### Genome-scale metabolic network reconstruction from genome sequences

We first filtered all genome sequences in the BV-BRC [46] 3.6.12 database to only include those that were considered "good" quality, "complete," and which came from "human" hosts.

BV-BRC guidelines define "good" as "a genome that is sufficiently complete (80%), with sufficiently low contamination (10%)", and amino acid sequences that are at least 87% consistent with known protein sequence. "Complete" means that replicons were completely assembled. "Human" hosts mean that the bacteria were isolated from a human host prior to sequencing. We recognize that the completeness cutoff is not incredibly stringent and has the potential to impact subsequent essential gene predictions. To address this limitation, we performed analyses which we present in S1 Text.

There are 538 species of bacterial pathogens [2], some of which either do not have publicly available genome sequences via BV-BRC or do not have "good" and "complete" genome sequences and were isolated from a human host in BV-BRC. For simplicity, we use the word pathogen throughout the manuscript to include bacterial species that are considered either pathogens or opportunistic pathogens. For all pathogens that pass the initial "good," "complete," and "human" filters, there is at least 1 NCBI taxID for each species, with some species having multiple unique NCBI taxIDs. Multiple genome sequences are available in BV-BRC for each NCBI taxID, so sequences were selected based on the presence of metadata in a hierarchical nature. This metadata requirement was instantiated because we wanted to select sequences with enough available metadata for downstream analyses. Sequences with the most associated metadata were prioritized. If multiple sequences had the same amount of metadata, we selected the sequence that had isolate environment-associated metadata. If multiple sequences fulfilled the previous requirements, the strain that had most health-associated metadata was selected. This hierarchical selection was continued for metadata categories of isolation country, collection date, and host age, in that order of priority. The resulting list contained 914 unique genome sequences. This procedure was automated with a python script available at https://github.com/emmamglass/PATHGENN.

All amino acid sequences were then automatically annotated with RAST 2.0 [47,48], and GENREs were created for each strain using the Reconstructor [17] algorithm. All models are publicly available (see Data Availability section). We benchmarked all GENREs using the community standard, MEMOTE [49], and have included overall MEMOTE scores and subcategory scores are reported in S2 and S3 Figs.

## Identifying core, accessory, and unique reactions and their corresponding metabolic subsystems

To identify core, accessory, and unique reactions across pathogens, we generated a reaction presence matrix. Rows corresponded to each individual GENRE, while columns were KEGG [50] reactions. Reaction presence and absence was noted for each genre (1 = presence, 0 = absence). Then, a histogram was generated based on frequency of reaction presence. Reactions that were present in less than 25% of GENREs were categorized as unique reactions, reactions present in 25% to 75% or GENREs were categorized as accessory reactions, and reactions present in greater than 75% or GENREs were categorized as core reactions. These cutoffs were selected based upon an analysis performed in a previously published study [15]. Subsequently, each reaction was annotated with the corresponding KEGG metabolic subsystem to which it belongs. The histogram was then annotated with these metabolic subsystems in each bar of the histogram. Secondly, we determined the proportion of reactions belonging to each metabolic subsystem in core reactions compared to unique reactions.

Subsequently, we wanted to ensure that there were indeed reactions present in only 1 strain by changing the number of bins in our histogram from 8 to 914. This analysis revealed that there were 232 reactions that were unique to only 1 strain in the collection. This analysis ensures that there are indeed reactions that meet the strictest criteria for uniqueness. We have included this analysis in S7 Fig.

## FBA and t-SNE dimensionality reduction/visualization

For each of the 914 models, Flux Balance Analysis (FBA) was performed using the COBRApy toolbox for each model in our collection to capture metabolic flux through all model reactions. The objective function used in our FBA analysis was biomass synthesis. This objective function was used because we use FBA to predict genes for biomass synthesis in subsequent analyses. Ten flux samples were taken per model for a total of 9,140 flux samples, since reducing the dimensionality of a larger number of flux samples was infeasible; t-SNE [51] was used for dimensionality reduction and subsequent visualization of the FBA output. Points were colored based on taxonomic class and subsequently colored on physiological location for visualization purposes. The perplexity parameter was selected to attempt to preserve local and global relationships in the data as best as possible, by using the relationship $= N^{\frac{1}{2}}$, where $P$ = perplexity and $N$ = number of points.

We chose to generate 10 flux samples per model due to computational limitations. More specifically, if we were to produce 500 flux samples per GENRE, this analysis would yield a dataset 457,000 rows × ~4,000 columns, which is computationally unfeasible to reduce the dimensionality using t-SNE. However, to ensure that 10 flux samples was sufficient to capture much of the flux solution space, we ran a subsequent t-SNE analysis. In this analysis, we randomly sampled 100 GENREs from the 914 total GENREs. Then, for each of those 100 GENREs we generated 100 flux samples for each GENRE and used t-SNE for dimensionality reduction and subsequent visualization (S8 Fig). We performed this analysis 4 times, each time selecting a different subset of 100 GENREs, to ensure that the results would hold true for multiple randomly selected subsets of GENREs. We observed similar clustering patterns with this larger sample of fluxes (100), in each of the 4 randomly selected subsets of 100 GENREs. Specifically, we still observe large clusters of *Gammaproteobacteria* and *Actinomycetia*. Additionally, we still observe the separation of *Epsilonproteobacteria* into distinct clusters, one of which is completely comprised of stomach isolates, suggesting that our original analysis using 10 flux samples per GENRE can capture variation in flux as well as using 100 flux samples.

Due to the high-dimensionality of our data set, linear dimensional reduction techniques like NMDS and PCA are not able to sufficiently display variation in the data set. To ensure that our data set could not be sufficiently visualized with NMDS and PCA, we performed the same analysis in Fig 3 using NMDS and PCA methods. The resulting plots are shown in S9 Fig. We observed that clustering in these plots is less defined. Because of less-defined clustering observed with linear dimensionality reduction methods, we concluded that the nonlinear dimensionality reduction method t-SNE was better for data visualization and testable hypothesis generation, despite losing the ability to draw conclusions using absolute distances between clusters.

## Determining uniquely essential genes

Essential genes for all 914 models were determined using an FBA-based single-gene-knockout method in COBRApy (cobra.flux_analysis.variability.find_essential_genes()). All essential genes were translated to KEGG orthologs. Strains and their corresponding essential genes were grouped by isolation site. Essential genes present in $> = 80\%$ of strains in a given isolation source were defined as uniquely essential to that isolation source. The most uniquely essential gene present in stomach isolates that was not considered uniquely essential to other isolation sites was selected, which were *thyX*.

## In vitro growth assay for computational prediction validation

We validated our computational prediction that *thyX* is uniquely essential to stomach isolates and can be targeted with lawsone. We selected 3 stomach isolates that were included in our

network reconstruction collection, *Arcobacter butzleri* (DSM 8739), *Helicobacter pylori* (DSM 21031), and *Campylobacter coli* (JV20). We selected 4 non-stomach associated isolates *Escherichia coli* (JM101), *Pseudomonas aeruginosa* (PAO1), *Porphyromonas gingivalis* (DSM 20709), and *Burkholderia cenocepacia* (K-56-2). Strain selection criteria are outlined in the Results.

We grew overnight cultures of each species prior to beginning each experiment. Each species was grown in a complete media to be consistent with the way computational predictions were done. However, different complete media were used for each species to optimize their growth capabilities. *A. butzleri*, *B. cenocepecia*, and *C. coli* were grown in Difco brain heart infusion broth (Becton, Dickinson & Co) supplemented with 5% FBS (gibco by Thermo Fisher Scientific). *H. pylori* was grown in brucella media (Remel) supplemented with 5% FBS. *E. coli* and *P. aeruginosa* were grown in Luria broth (Sigma). *P. gingivalis* was grown anaerobically in reinforced clostridial media (ATCC medium 2107). *A. butzleri*, *H. pylori*, and *C. coli* are micro-aerophilic species, so they were grown in an airtight container with a Mitsubishi Anaeropak to keep the oxygen concentration between 6% and 12% and carbon dioxide between 5% and 8%. All species were grown at 37˚C with the exception of *A. butlzeri*, which was grown at 28˚C.

Initial strong solutions of cerulenin (Sigma), $\alpha$-mangostin (MedChemExpress), and lawsone (2-Hydroxy-1,4-napthoquinione, Sigma) inhibitors were created by first using dimethyl sulfoxide (Sigma) to solubilize each compound. Brain heart infusion media was used for subsequent dilutions to achieve the necessary final concentration.

First, we performed a minimum inhibitory concentration (MIC) assay to determine the MIC of each compound for *A. butzleri*, one of the stomach isolates. Beginning with a high concentration of each compound, we plated 2× serial dilutions of each compound with *A. butzleri* from an overnight culture. We then ran a continuous growth curve using the Cerillo Stratus plate reader, encased in an air-tight container with a Mitsubishi anaeropak to achieve the microaerophilic conditions necessary for *A. butzleri* growth. Results of the MIC assay for each compound are shown in S10 Fig. The resulting MIC for cerulenin, $\alpha$-mangostin, and lawsone are 2 mM, 34.3 μm, and 320 M, respectively.

The resulting MIC of each compound for *A. butzleri* was used in our final validation assay. For this assay, we subjected each species to the same concentration of inhibitors 2 mM, 34.3 μm, and 320 μm for cerulenin, $\alpha$-mangostin, and lawsone, respectively, in a 96-well plate. Each condition had 8 wells containing the compound and bacteria, 8 wells with the bacteria and media, and 8 blank wells containing media. One plate was used per bacterial species to ensure no contamination occurred. After inoculating the plate, the plate was sealed with a Breathe-easy film to ensure gas exchange. The plate was then placed into the Cerillo stratus plate reader and monitored through stationary phase. Growth curve data was then downloaded from the plate reader via the Cerillo Canopy and saved on a local machine for analysis.

## Supporting information

**S1 Text. Completeness cutoff does not impact essentiality predictions.** This document includes 2 additional analyses to support the idea that the 80% completeness threshold does not impact our gene essentiality predictions. The data underlying Figs 1 and 2 is S1 Text can be found in Data1.zip.
(PDF)

**S1 Fig. Development of the PATHGENN GENRE collection.** The BV-BRC database was used to select pathogen genome strains that satisfied quality criteria. These genome strains were then annotated using the RAST annotation toolbox to generate the amino acid FASTA file that was then used in Reconstructor to generate the 914 GENREs in the collection.
(PNG)

**S2 Fig. Overall and subcategory MEMOTE scores across the GENRE collection.** Overall MEMOTE scores had an average of 84%; subcategory scores were also considerably high with minimal variability in quality. The data underlying S2 Fig can be found in Data2Data3.csv on Zenodo: https://zenodo.org/records/13952471.
(PNG)

**S3 Fig. Overall and subcategory MEMOTE score statistics.** Average, standard deviation, median, minimum, and maximum reported for overall MEMOTE score as well as the consistency, metabolite annotation, reaction annotation, gene annotation, and SBO annotation subcategories. The data underlying S3 Fig can be found in Data2Data3.csv on Zenodo at DOI: https://zenodo.org/records/13952471.
(PNG)

**S4 Fig. Summary of gapfilled reactions and metabolites across the 914 GENREs.** (a) Percent of total reactions gapfilled, (b) percent of total metabolites gapfilled. The data underlying S4 Fig can be found in Data4.xlsx on Zenodo: https://zenodo.org/records/13952471.
(PNG)

**S5 Fig. Phylogenetic tree of oral Fusobacteria and Bacteroidia species and Epsilonproteobacteria species in PATHGENN with annotated 16s rRNA sequences.** Fusobacteria and Bacteroidia species in the oral environment are not genetically similar. Epsilonproteobacteria are genetically similar, but occupy distinct environments. The data underlying S5 Fig can be found in Data5.txt on Zenodo: https://zenodo.org/records/13952471.
(PNG)

**S6 Fig. Commensal essential gene analysis.** List of essential genes in 4 commensal bacterial species (listed by species name and BV-BRC genome ID). Essential genes are reported in KEGG identifiers.
(PNG)

**S7 Fig. Histogram of annotated reactions across models.** This plot is supplemental to the data presented in Fig 2A, using more stringent unique reaction cutoffs. The histogram has 914 bins which allow us to see that 232 reactions are unique to one GENRE. The data underlying S7 Fig can be found in Data7.csv on Zenodo: https://zenodo.org/records/13952471.
(PNG)

**S8 Fig. t-SNE plot of 100 flux samples for 100 GENREs.** The clustering relationships seen in Fig 4 with 10 flux samples for each of 914 models are consistent with the clusters seen here with 3 randomly selected subsets of 100 GENREs with 100 flux samples each. Each pair of plots (a and e, b and f, c and g, d and h) represents randomly selected subset of 100 GENREs; (a–d) are colored based on taxonomic class; (e–h) are colored based on physiological location. The data underlying S8 Fig can be found in FluxSampleData.zip on Zenodo: https://zenodo.org/records/13952471.
(PNG)

**S9 Fig. Comparison of principal component analysis and non-metric multidimensional scaling method on flux data.** (a) principal component analysis, colored on taxonomic class. (b) Non-metric multidimensional scaling colored on taxonomic class. (c) Principal component analysis colored on isolate physiological location. (d) Non-metric multidimensional scaling. The data underlying S9 Fig can be found in FluxSampleData.zip on Zenodo: https://

zenodo.org/records/13952471.
(PNG)

**S10 Fig. Results from MIC assay.** MIC assay with *Arcobacter butzleri* for each chemical inhibitor. Stars indicate the selected MIC, the concentration used in the subsequent validation experiments. The data underlying S10 Fig can be found in Data10.zip on Zenodo: https://zenodo.org/records/13952471.
(PNG)

## Author Contributions

**Conceptualization:** Emma M. Glass, Lillian R. Dillard, Jason A. Papin.

**Data curation:** Emma M. Glass.

**Formal analysis:** Emma M. Glass.

**Funding acquisition:** Emma M. Glass, Lillian R. Dillard, Jason A. Papin.

**Investigation:** Emma M. Glass.

**Methodology:** Emma M. Glass, Lillian R. Dillard, Glynis L. Kolling.

**Project administration:** Emma M. Glass.

**Resources:** Emma M. Glass.

**Software:** Emma M. Glass, Lillian R. Dillard, Glynis L. Kolling, Andrew S. Warren.

**Supervision:** Glynis L. Kolling, Jason A. Papin.

**Validation:** Emma M. Glass.

**Visualization:** Emma M. Glass.

**Writing – original draft:** Emma M. Glass.

**Writing – review & editing:** Emma M. Glass, Lillian R. Dillard, Glynis L. Kolling, Andrew S. Warren, Jason A. Papin.

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
