## [Editor Report · Decision Letter 0]

28 Mar 2024

Dear Jason, 

Thank you for submitting your manuscript entitled "Evolution shapes metabolic function and niche-specific antimicrobial targets in pathobionts" for consideration as a Research Article by PLOS Biology.

Your manuscript has now been evaluated by the PLOS Biology editorial staff, and I'm writing to let you know that we would like to send your submission out for external peer review.

Once your full submission is complete, your paper will undergo a series of checks in preparation for peer review. After your manuscript has passed the checks it will be sent out for review. To provide the metadata for your submission, please Login to Editorial Manager (https://www.editorialmanager.com/pbiology) within two working days, i.e. by Apr 01 2024 11:59PM.

Kind regards,

Roli

Roland Roberts, PhD

Senior Editor

PLOS Biology

rroberts@plos.org

---

## [Decision Letter · Decision Letter 1]

4 Jun 2024

Dear Jason,

Thank you for your patience while your manuscript "Evolution shapes metabolic function and niche-specific antimicrobial targets in pathobionts" was peer-reviewed at PLOS Biology. It has now been evaluated by the PLOS Biology editors, an Academic Editor with relevant expertise, and by three independent reviewers. 

You'll see that all three reviewers are positive, but each raises a number of concerns that will need to be addressed. For example, reviewer #1 notes the need to have tighter completeness and contamination cutoffs, and requires methodological detail of the flux balance analysis, and more detail on (and/or better) experimental validation. Reviewer #2 has a long list of requests. As far as I can see, the first point requires significant extra work (in silico KO of “uniquely essential” genes in commensals); point 8 also, potentially (re-doing the tSNEs with PCoA etc) – the remainder are requests for clarification, discussion, etc. Most of reviewer #3's requests are for improving the clarity of the manuscript and the accuracy of the claims, and can be addressed textually, but there’s a potential experimental request relating to testing of the non-stomach pathobionts.

IMPORTANT: I discussed these comments with the Academic Editor, who generally found all of the reviewers' comments to be reasonable, but particularly wanted me to emphasise the following:

- Points 1 and 2 of reviewer #1. The absence of 16S regions from most genomes is quite confusing and should be clarified. Point 4 of reviewer #2 is closely related.

- Points 1 and 8 of reviewer #2. I am not quite sure how much work this is, but it should certainly be feasible for these authors.

- Point 2 of reviewer #2. Using simpler dimensionality reduction techniques for comparison would be helpful and should not be a problem.

- Point 1 of reviewer #3. I had similar problems when I first read the title and abstract. It would be good to streamline the writing, shorten and improve the structure of the abstract, and at least reconsider the use of "evolution" in the title.

In light of the reviews, which you will find at the end of this email, we would like to invite you to revise the work to thoroughly address the reviewers' reports.

Given the extent of revision needed, we cannot make a decision about publication until we have seen the revised manuscript and your response to the reviewers' comments. Your revised manuscript is likely to be sent for further evaluation by all or a subset of the reviewers.

**IMPORTANT - SUBMITTING YOUR REVISION**

*Re-submission Checklist*

*Published Peer Review*

*PLOS Data Policy*

*Blot and Gel Data Policy*

Sincerely,

Roli

Roland Roberts, PhD

Senior Editor

PLOS Biology

rroberts@plos.org

REVIEWERS' COMMENTS:

Reviewer #1:

The manuscript by Glass et al. reports am interesting approach to antimicrobial target identification using genome-scale metabolic modelling. The central idea is to identify essential genes in a group of microbes that occupy the same "niche" under the infection context. Metabolic modelling provides an in-silico screen and enumeration of essential genes, with a subset of results validated in vitro using known small-molecule inhibitors. Overall, the manuscript is a valuable addition as a resource of metabolic models of pathogens with a proof-of-concept analysis showing how the resource could be used. There are, however, several points where the study is unclear / falls short of providing sufficient evidence to the conclusions. These are listed below, and I hope that these will be useful for the authors to improve the study/presentation.

1. Genome selection for reconstruction: 80% completeness and 20% contamination are too loose cut-offs as these can introduce both type I and type II errors. This is especially critical since the subsequent analysis is categorical, based on presence/absence of genes (and mapped reactions thereby). The importance of this point is highlighted by the fact that a vast number of the genomes did not have 16S regions.

2. Details of FBA regarding Objective function etc are missing from the Methods; a discussion on the choice of objective function for identifying flux distributions is important as in vivo objective functions may include more metabolic tasks than growth. Also unclear how flux variability was accounted for - 10 simulations are far too less for this. Was pFBA used?

3. Experimental validation: how were the four non-stomach bugs selected? Ideally a broader range of microbes would be useful to assess specificity of the targets. If less, at least some phylogenetic control should be applied.

4. Literature on conditional essentiality should be included in the discussion for giving a broader context to the study as it currently does assume an arbitrary complete growth media for all species.

Reviewer #2:

The study by Glass et al. investigates evolutionary principles that may have shaped niche-specific metabolic functions in human bacterial pathogens, and applies these insights to identify antimicrobial targets specific to those niches. Utilizing genome-scale metabolic network reconstructions (GENREs) for known human bacterial pathogens from the BV-BRC database, the researchers identified physiological niche as a key factor in determining metabolic function. Through their in silico analysis, the researchers demonstrated the discovery of unique essential genes in stomach-associated pathobionts, and targeted these genes with small-molecule inhibitors. In vitro validation showed that inhibiting genes such as fabZ, fabF, and thyX selectively impeded the growth of stomach-specific pathobionts. This research highlights the potential of niche-specific antimicrobial strategies to mitigate the widespread effects of antibiotics and tackle the global antimicrobial resistance (AMR) crisis.

This work could be of great interest to the AMR community, as it advances our understanding of pathobiont metabolism and pioneers the development of targeted antimicrobial therapies based on metabolic and environmental specificities. The manuscript is well-crafted, and the display items effectively communicate the high quality of the research presented. However, to enhance the current version of the manuscript, I offer the following major and minor comments for consideration.

Major comments:

1. A central theme of this work is the importance of niche-specific metabolic signatures identified through genome-scale metabolic pathway analysis. This approach, if successful, could mitigate the reliance on broad-spectrum antibiotics. However, there is a significant concern regarding the impact on commensal microbes. As the manuscript notes, phylogenetically distinct bacteria can exhibit similar metabolic functions due to convergent evolution. Therefore, *niche-specific* antimicrobials might still produce *broad-spectrum* effects within the localized physiological area, which is particularly concerning for gut microbes given their critical role in human health. This potential issue is the Achilles heel of the study and should be addressed in the manuscript. Additionally, I recommend that the authors conduct a proof-of-concept analysis to demonstrate that in silico knockouts of the three uniquely essential genes do not adversely affect the metabolic growth of a variety of stomach commensals.

2. The study emphasizes the role of environmental selection pressure as a driving force behind both divergent and convergent evolution. While the application of computational and empirical methods to explore these evolutionary dynamics is indeed innovative, the conceptual framework presented is not new. Evolutionary biology has long discussed these principles. Therefore, it is crucial that the authors more thoroughly integrate and discuss existing literature on this topic. Acknowledging prior foundational studies will provide a clearer backdrop against which the novelty of their own methodological and computational approaches can be distinguished. This could involve a more detailed comparison of their results with those of previous studies, particularly emphasizing any novel implications for antimicrobial therapy development.

3. The manuscript makes significant use of the MEMOTE score to assess the quality of metabolic network reconstructions, reporting an average score of 83%. For readers not familiar with genome-scale metabolic reconstruction, it would be beneficial if the authors could include a brief explanation of how the MEMOTE score is quantitatively defined. (This reviewer acknowledges the qualitative description in the Results section.) Clarification on whether an average score of 83% is indicative of high quality would be particularly helpful. Additionally, the assertion that this score is "…suggesting all reconstructions in the collection are of high quality…" is misleading. Given the variability inherent in any biological data set, it is likely there are some outliers with lower scores.

4. The authors utilize 16s rRNA sequences to define genetic similarity among bacterial strains, limiting the analysis to 362 sequences out of the 914 strains due to the availability of 16s rRNA data. This methodological choice raises a concern about the comprehensiveness and accuracy of genetic similarity assessments within the study. Considering the availability of complete genome sequences for many microbial strains, using whole-genome data could provide a more robust and inclusive measure of genetic similarity. Without asking to re-do the work, I recommend that the authors discuss the rationale behind relying solely on 16s rRNA sequences.

5. The statement that "these results underscore the potential of environment to shape unique metabolic function, and more importantly, show that the metabolic network reconstructions in our collection can capture complex evolutionary dynamics through constraint-based analysis" seems to overreach the conclusions that can be drawn from the data presented. The analysis described in Figure 2, which appears to rely on gene annotation analysis or gene functional enrichment, does not necessarily require the use of metabolic network reconstructions. Such analyses could potentially be conducted without the sophisticated framework of GENREs, using more straightforward bioinformatic tools. I suggest that the authors either clarify how the metabolic network reconstructions uniquely contribute to capturing complex evolutionary dynamics in ways that conventional gene annotation or enrichment analysis cannot, or consider revising the statement to more accurately reflect the methodology and its impact.

6. The use of "flux samples" in the study raises several questions that are not addressed in the manuscript. Flux sampling is inherently dependent on the input conditions, such as the composition of the growth media. This dependency can significantly influence the metabolic state modeled by the GENREs. It is crucial for the authors to specify the media conditions used during these simulations, as this would impact the generalizability and applicability of the findings. Importantly, how do the authors know that their media conditions are physiologically relevant, especially in the case of the stomach? Additionally, the decision to use 10 flux samples per model rather than a single sample or another number should be justified. Moreover, it would be beneficial if the authors could explain how these 10 samples were differentiated from each other within the same model. Were there variations in the initial conditions, or were different constraints applied to each sample?

7. Given that t-SNE includes a stochastic element and can sometimes exaggerate the identification of clusters depending on the chosen parameters, it would be useful for the authors to justify their preference for this method over more traditional and straightforward techniques such as Principal Coordinate Analysis (PCoA) or Multi-Dimensional Scaling (MDS). These alternatives are not only more commonly used in similar studies, but also tend to provide a more interpretable and reliable framework for understanding high-dimensional data. While the conclusions drawn by the authors may not heavily depend on the dimensionality reduction technique used, I am curious to see whether the same patterns and clusters observed using t-SNE would also be evident with these alternative approaches, which could further validate the findings and provide a more comprehensive understanding of the data.

8. "Additionally, targeting to the infection site could reduce the need for bacterial community characterization since all organisms in the environment would be targeted." This echoes my point #1 above. Do the authors suggest that the eradication of all commensals in the niche might be acceptable, or is this an oversight? How does this strategy fundamentally differ from the problems associated with the use of broad-spectrum antibiotics, which also disrupt healthy microbial communities?

9. It would be helpful to the reader if the authors could discuss limitations of the work and areas for future research, as is customary towards the end of the Discussion section in many high-profile papers.

10. The definition of "unique reactions" used in the manuscript, set at a threshold where reactions present in 25% or fewer of the pathobionts are considered unique, appears overly inclusive. To this reviewer, reactions that are truly unique would be those found exclusively within a single class, genus, or species. Could the authors simply clarify whether any reactions met this stricter criterion of uniqueness? If so, detailing these findings could significantly strengthen the argument for the novelty and specificity of the metabolic functions uncovered in the study.

11. I found that there was a time delay in seeing the effects of alpha-mangostin on C. coli. Could the authors provide insights or hypotheses regarding this delayed response?

Minor comments:

1. Please add a space in "pathobionts(Figure 3d)". Also, please make sure that spaces are correctly placed before reference annotations, e.g., "for this application(29)".

2. Please remove the unnecessary space in "bacteria (28) ." (right before the period.)

3. In the Methods section, please change "taxid" to "taxID".

4. In the legend of Figure 1, please correct "1.15 metabolites" to "1.15 million metabolites".

5. The designations in the legend of Figure 1 are incorrect. For instance, "(d) The relationship between the number of genes and the number of reactions..." should be labeled as (g). Please revise accordingly.

6. Please correct the spelling of "uniuqley" in the caption for Table 1 to "uniquely".

7. In the legend of Figure 2, it states the results in (b) are an enrichment analysis. This should be clarified as it is not a statistical enrichment but simply reporting proportions.

8. In Figure S1, the instruction "Select one strain per NCBI TaxID, strain with most pertinent metadata" is unclear. Consider revising the punctuation for clarity.

9. For consistency, the title of Figure 3's legend should not be in title case. Also, add a space in "location(b)".

10. The legend for Figure 4 needs significantly more detail. For example, explain what the + and - symbols represent, the meaning behind the colors (green, yellow, orange) of these symbols, and how the confidence intervals (for the lines in the growth curves) are defined. Additionally, change "α - mangostin" to "α-mangostin" (replace the en dash with a hyphen; remove unnecessary spaces).

Reviewer #3:

The manuscript "Evolution shapes metabolic function and niche-specific antimicrobial targets in pathobionts" reports the development of constraint-based genome scale modelling approach to identify antimicrobial targets unique to specific microbial niches rather than to bacterial species. The authors construct and analyse the metabolic networks of >900 pathobionts (>300 species) and identify evolutionary principles guiding the development of unique niche-associated metabolic signatures. They further focus on stomach pathobionts, and identify and validate three genes unique essential to isolates of this particular physiological niche.

The topic of the manuscript is certainly relevant, the approach is innovative and the reconstructed network compendium is a valuable resource for the community. Therefore, I am generally enthusiastic about this story. However, I do have few issues that I think should be addressed in order to solidify the findings.

Major comments

I think the manuscript would vastly benefit from streamlining to improve readability. In particular:

1. the subtitles within the results section do not always accurately reflect the findings. Or at least it is not obvious from the content of the text how to make it to the title. This also applies to the manuscript title. While the authors suggest evolutionary relationships throughout the manuscript, (not being an evolutionary biologist) in my view none of them is adequately tested/verified/substantiated in sufficient detail. The authors mostly suggest/predict such relationships. Thus, I advise caution in over-using them term "evolution", even in the tittle. Increasing accuracy of the title and subtitles towards what each section actually describes would likely solve the issue to a great extent.

2. explaining and interpreting the findings from Figs. 2c and 3a & b should be more precise, less speculative and better harmonized. The authors tend to very quickly dive into speculative interpretation, but do not provide accurate definitions of specific terms, neither description of general observations to be derived from the plots. The paragraphs below provide some specific examples (not exhaustive), as well as a few suggestions on how I think one could improve the issues. 

The section "Evolutionary drivers of unique metabolic function in pathobionts" and how it relates with Fig 2b is very difficult to follow. The message to be passed here is not clear/diluted among rather superficial/possible interpretations. I suggest a couple of possible improvements:

* Provide a more precise definition of "metabolic niche similarity". What is the meaning of "pairwise"? Improve the annotation of the axes in Fig 2c to overlap with text and ease readability of figure-text.

* The overall guidance for interpretation of Fig 2c could be facilitated by stating what is actually observed before trying to interpret it. For instance, what I first see from this plot is that similarity between essential genes is primarily higher than, and generally not correlated with, genetic similarity (assuming that I understand what is plotted). So it is somewhat unnatural that the first statement about this plot focus on "strong correlation". I total agree that this strong correlation is there - exceptionally for high similarity pathobionts, but it takes an eye to get there, especially if one is not familiar to the topic/representation.

* While both variables don't seem to generally correlate, they do show a general trend, even well captured by line of best fit (which is otherwise not really used. Why get it then?). While the authors equally highlight 4 different potential examples across the large cloud, it is unclear how they decided to subset these 4 different groups. Most importantly, the quantitative aspect that group 3 is clearly breaking (here I would dare say evolutionary) tendency, and potentially giving the most interesting outcome, is completely missed/not mentioned. To my understanding, this is the take-home-message of this plot: group 3 has a much higher pairwise essential gene similarity than what can be predicted from genetic similarity across the >900 GENREs. So I would be interested in knowing more about which pathobionts/genes are in fact grouping in this cloud - and whether they potentially have similar de facto metabolic niches, rather than a long list of examples sampling subsets 1, 2 and 4 - which seem to follow the general trend anyway.

* I would suggest a more descriptive tittle of this section. The current tittle does not really describe what is done/shown, and can mislead the reader towards the wrong expectation.

* The findings of this sections are disconnected from the remaining manuscript.

Similarly to Figure 2c, the guidance provided for interpretation of Fig. 3 could be improved. For instance, the authors state "We observed several clusters of interest in these plots, which we considered more deeply for theoretical analysis." This might mislead the reader to think that both taxonomic class & physiological environment equally sustain the grouping of metabolic phenotypes. However, when looking at the figure, taxonomic cluster seems to be much more prominent than physiological environment - even though the second also seems to be there to a less extent. Also here, it would help streamlining the massage to make it more clear. Important: what the authors mean with "in silico metabolic phenotypes" is again not clearly defined in the main text, and this hinders following interpretation.

It is unclear how the authors selected the species/strains for testing drug activity. The three non-stomach pathobionts are taxonomically so different from the stomach pathobionts, that one cannot be sure what actually drives drug activity - for instance drug transport or other intrinsic resistance mechanisms. I also miss the point the authors want to make: strain/species specificity or environmental niche specificity? Would it be more helpful to test taxonomically more similar strains (stomach and non-stomach) across different culturing medium - stomach-like vs non-stomach like? If at all possible. This would also support the manuscript towards establishing what is novel regarding the three selected drugs. Right now quite confusing and some text parts are redundant.

Minor comments

Abstract

Abstract could be a bit more streamlined. Some sentences - especially in the beginning - come unconnected. In addition, the third sentence doesn't read well - check grammar and revise. 

Results section

Far stretch to understand what the author mean with "This logarithmic trend is consistent with the expectation that there are limited evolutionary advantages for bacteria with increasingly large genomes (17), which further validates the relevance of our collection." A more explicit explanation would be great.

"We can analyze metabolic functionality through the generation of flux distributions feasible in a network reconstruction". This sentence doesn't read well - check grammar and revise. 

I generally advice caution in using the word "enriched" and "significant" without a statistical test. For instance while describing the results of Fig 2b, both words are mentioned, but it is unclear to which null hypothesis the authors refer to.

Fig. 2:

Panel b could be re-ordered according to enrichment core vs unique to make it easier to find the different metabolic functions mentioned in the text.

Panels b and c: b and c are inside the plots. Looks strange.

---

## [Decision Letter · Decision Letter 2]

4 Oct 2024

Dear Jason,

Thank you for your patience while we considered your revised manuscript "Physiological niche-specific metabolic function and associated antimicrobial targets in pathobionts" for consideration as a Research Article at PLOS Biology. Your revised study has now been evaluated by the PLOS Biology editors, the Academic Editor, and the original reviewers.

You'll see that reviewer #1 says s/he’s puzzled by your response to one of their previous queries, and emphasises some of the ways that it might be addressed. Reviewer #2 is very positive and just has some textual and presentational requests. Reviewer #3 is mostly positive, but has difficulties with your "Pathobiont metabolic niche analysis supports key evolutionary ideas" section, which s/he thinks is hard to follow and not well supported.

In light of the reviews, which you will find at the end of this email, we are pleased to offer you the opportunity to address the remaining points from the reviewers in a revision that we anticipate should not take you very long. We will then assess your revised manuscript and your response to the reviewers' comments with our Academic Editor aiming to avoid further rounds of peer-review, although might need to consult with the reviewers, depending on the nature of the revisions.

**IMPORTANT - SUBMITTING YOUR REVISION**

*Resubmission Checklist*

*Published Peer Review*

*PLOS Data Policy*

*Blot and Gel Data Policy*

Sincerely,

Roli

Roland Roberts, PhD

Senior Editor

PLOS Biology

rroberts@plos.org

REVIEWERS' COMMENTS:

Reviewer #1:

I am puzzled by the authors' response to the completeness and contamination issue. The missing 16S comment was to highlight that essential genes can be missed. the "good quality" category used in genomic analyses have a different purpose and thus should not be used prima facie for essentiality analyses. I do recognise limitations in addressing this issue though but below are a few ways that I would like to suggest:

1. Select a subset of higher quality (say>95% complete) genomes and see if consistent results are found

2. Compare the results with known universally essential genes as +ve controls

3. add a discussion clearly identifying this limitation

Reviewer #2:

The authors have done a terrific job in addressing my extensive comments. I have no major comments to add at this point. I have only minor comments at this point to improve the readability of the manuscript.

Minor comments:

- Spacing before references remains inconsistent throughout the paper. Please see [1] and [2], for instance, and others.

- Figure 1c is not mentioned in the manuscript.

- Please add a period after the last sentence in the legend of Figure 1.

- Is reference 23 on line 136 necessary?

- "Consistent with previous studies, we observe clustering based on taxonomic class [20,31,32] (Figure 3a)." It would be proper to put the references right after "previous studies" and not at the end of the sentence where you've described your finding.

- Line 311: "...organisms do not utilize. [35]." Please use one period for each sentence.

- Please see the legend for Figure 3. "Areas of interest from (a) and (b) are highlighted if (c) - (f) and described in the main text." This description is confusing to me. If c-f are described (and they are), then you've highlighted (a) and (b)?

- I apologize, but I am very confused by the structure of Table 1. What is the blank space on the lower right corner? Why are there eight genes in the table and not seven? Should fabF be in this table? It would be great if the co-authors could help make the table more visually appealing and better organized.

- Line 360: Is there a reason thyX needs to be underlined?

- The authors mention Figure 4a through 4u in the manuscript, but these indications are not present in the actual figure nor mentioned in the legend. Also, "i" and "j" are not mentioned in the main text.

Reviewer #3:

First, I appreciate the efforts of the authors to satisfy the reviewer requests. Many of the requests have been addressed, especially to better document the quality of the dataset (e.g. contextualization of MEMOTE scores) and improve definitions used throughput the manuscript (e.g. metabolic niches). The precision and accuracy of the abstract, subtitles and general text has improved, and thus readability is better. I also appreciate the improvement and extra clarifications of the last section, and understand the limitations do expand on experimental efforts.

Nevertheless, in my opinion, the section "Pathobiont metabolic niche analysis supports key evolutionary ideas" remains very difficult to follow: difficult terminology, lack of precise examples from own data and poorly integrated in the rest of the manuscript. The authors start with phenotypic evolution (which I don't really understand the meaning of), but never come back to that over the course of the section. While I appreciate the description of general tendency and clarification of the outlier group, many of my initial concerns remain: on what statistical basis do the authors distinguish sections 3 and 4 from the general tendency? Their interpretation of these two sections is based on general examples - none of the data that is actually at hand and part of the sections - and arguments and conclusions are made based on physiological locations, which are not part of this analysis and comes afterwards in figure 3. Also: Figure 2c is not mentioned early enough in the revised text, should be in line 206. Finally, the authors mention in section "The identification of uniquely essential genes informs stomach pathobiont-specific antimicrobial targets and growth inhibitors" starting at line 330: "To identify uniquely essential genes to stomach-associated pathobionts in silico, we first determined essential genes for all strains in the GENRE collection using an FBA single-gene-knockout method in COBRApy. Subsequently, we instantiated a universal essentiality threshold to determine uniquely essential genes across stomach-associated pathobionts (see Methods)." Why did they do this all again? Is it not what was done for Fig. 2c to determine metabolic niche similarity? It is my opinion that this section is particularly weak and, as is, not needed to support the rest of the manuscript.

Minor comments

Line 11: Pathobiont is used in abstract prior to definition.

Line 57: Reference missing?

Lines 95-97: Sentence does not make sense. Suggest revise.

Lines 125-127: While this attempts to give a reference to the reader, it comes out of context, and therefore does not really help. What is ModelSEED and CarveMe?

Line 144: I totally understand the concept of gapfilling, and thus I am aligned with the authors that it can add value to the networks beyond gene annotation limitations. However, this argument is vague/weak because it is not objective. How many reactions were added with gapfilling? Even if not many, are they essential reactions? Objectivity here would strengthen the argument.

Lines 160 & 171: The start of these paragraphs is almost the same, but see to contradict each other:

"Most reactions across pathobionts were considered unique, which can be attributed to the large taxonomic range of pathobionts in the GENRE collection."

"The majority of reactions across pathobionts were considered unique to less than 25% of pathobionts showing that subgroups of pathobionts share these unique reactions"

Can the authors somehow illustrate what they want to say in the second paragraph? Otherwise it is difficult to follow.

Line 388: Respectfully, Gram-negatives generally have a cell wall!!! Gram-staining pertains to outer-membrane and not cell wall. The argument is correct though, because the outer-membrane is indeed a major barrier for antimicrobials in Gram-negatives. Please correct ;)

Discussion: Can/should be streamlined, as at places the language can be improved.

---

## [Editor Report · Decision Letter 3]

17 Oct 2024

Dear Jason,

Thank you for your patience while we considered your revised manuscript "Physiological niche-specific metabolic function and associated antimicrobial targets in pathobionts" for publication as a Research Article at PLOS Biology. This revised version of your manuscript has been evaluated by the PLOS Biology editors and the Academic Editor.

Based on our Academic Editor's assessment of your revision, we are likely to accept this manuscript for publication, provided you satisfactorily address the following data and other policy-related requests:

IMPORTANT - Please attend to the following:

a) We wonder if you could make your Title more explicit and accessible (and declarative). Specifically, we suggest changing it to "Niche-specific metabolic phenotypes can be used to identify antimicrobial targets in pathobionts" - happy to discuss this by email.

b) Relatedly, we're wondering how wedded you are to "pathobiont"? I note that one of the reviewers already asked you to define this, and I'm uncertain how widespread this term is. To what extent is your study aimed at *pathobionts* rather than *pathogens*?

c) Please address my Data Policy requests below; specifically, we need you to supply the numerical values underlying Figs 1ADEFG, 2AB, 3ABCDEF, 4, S2, S4AB, S5 (treefile), S7, S8ABCDEFGH, 9ABCD, 10ABCD, and Figs 1 & 2 in File S11, either as a supplementary data file or as a permanent DOI’d deposition. I note that you already have an associated GitHub deposition, but this is currently only contains code and raw data. Please could you complete this deposition with the data underlying the Figures? IMPORTANT: Also, because Github depositions can be readily changed or deleted, please make a permanent DOI’d copy (e.g. in Zenodo) and provide this URL (see below).

d) Please cite the location of the data clearly in all relevant main and supplementary Figure legends, e.g. “The data underlying this Figure can be found in S1 Data” or “The data underlying this Figure can be found in https://zenodo.org/records/XXXXXXXX

e) I note that many of the supplementary files seem unnecessarily large, and should be cropped to reduce file size. "Figs" S3 and S6 would be better provided as Tables (spreadsheets) rather than these very large Tiffs.

f) Please make any custom code available, either as a supplementary file or as part of your data deposition. I think this is probably already in your Github/Zenodo deposition, but I'm just checking.

We expect to receive your revised manuscript within two weeks. 

*Published Peer Review History*

*Press*

Sincerely,

Roli

Roland Roberts, PhD

Senior Editor

rroberts@plos.org

PLOS Biology

DATA POLICY:

Regardless of the method selected, please ensure that you provide the individual numerical values that underlie the summary data displayed in the following figure panels as they are essential for readers to assess your analysis and to reproduce it: Figs 1ADEFG, 2AB, 3ABCDEF, 4, S2, S4AB, S5 (treefile), S7, S8ABCDEFGH, 9ABCD, 10ABCD, and Figs 1 & 2 in File S11. NOTE: the numerical data provided should include all replicates AND the way in which the plotted mean and errors were derived (it should not present only the mean/average values).

CODE POLICY

DATA NOT SHOWN?

---

## [Editor Report · Decision Letter 4]

21 Oct 2024

Dear Jason,

Thank you for the submission of your revised Research Article "Niche-specific metabolic phenotypes can be used to identify antimicrobial targets in pathogens" for publication in PLOS Biology. On behalf of my colleagues and the Academic Editor, Tobias Bollenbach, I'm pleased to say that we can in principle accept your manuscript for publication, provided you address any remaining formatting and reporting issues. These will be detailed in an email you should receive within 2-3 business days from our colleagues in the journal operations team; no action is required from you until then. Please note that we will not be able to formally accept your manuscript and schedule it for publication until you have completed any requested changes.

Sincerely, 

Roli

Senior Editor

PLOS Biology

rroberts@plos.org